# Inhibition Effect of Non-Host Plant Volatile Extracts on Reproductive Behaviors in the Diamondback Moth *Plutella xylostella* (Linnaeus)

**DOI:** 10.3390/insects15040227

**Published:** 2024-03-26

**Authors:** Junxiang Zhou, Zhen Zhang, Haotian Liu, Mengbo Guo, Jianyu Deng

**Affiliations:** Department of Plant Protection, Advanced College of Agricultural Sciences, Key Laboratory of Quality and Safety Control for Subtropical Fruit and Vegetable, Ministry of Agriculture and Rural Affairs, Zhejiang A&F University, Hangzhou 311300, China; isakzjx@foxmail.com (J.Z.); zzbrant@foxmail.com (Z.Z.); lockhart2024@163.com (H.L.)

**Keywords:** *Plutella xylostella*, pest behavioral control, essential oil, inhibition effect

## Abstract

**Simple Summary:**

Essential oils (EOs) from many non-host plants have been reported to have insecticidal and antifeeding activities to larvae and oviposition deterrent effects on the female moths of *Plutella xylostella*. However, their effect on sex pheromone communication during mate location has been less studied. Here, we studied the antennal response of both adult sexes to seven non-host plant EOs and their inhibition effect on the sex pheromone orientation of males and the oviposition of female moths. The results demonstrated that 10 mg of calamus (*Acorus gramineus*) and citronella (*Cymbopogon citratus*) EOs reduced the attraction of synthetic sex pheromones to male moths up to 72% and 66% in a sensitive way. The calamus EO also decreased the egg-laying number of female moths on host plants, with the highest inhibition rate of 100%. These findings contribute to a better understanding of the roles of volatile plant secondary metabolites in modulating reproductive-related behaviors, and exploit EOs and plant resources that can be used for the behavioral control of *P. xylostella*.

**Abstract:**

The pest management of *Plutella xylostella*, the global pest of cruciferous plants, is primarily dependent upon continued applications of insecticides, which has led to severe insecticide resistance and a series of ecological concerns. The essential oils (EOs) of non-host plants are considered to have a high application potential in pest behavioral control. In *P. xylostella*, the insecticidal properties, antifeeding activities, and oviposition inhibition effects of many EOs have been studied in larvae and female moths. However, less focus has been placed on the inhibitory effect on sex pheromone communication during courtship, which is vital for the reproduction of the offspring. In this study, by combining electrophysiological studies, laboratory behavioral assays, and field traps, we demonstrated that non-host plant EOs significantly inhibited the reproductive behaviors of both sexes. Notably, the calamus (*Acorus gramineus*) EO inhibited the preference of male moths for synthetic sex pheromone blends and reduced the egg-laying number of female moths on host plants, with the highest inhibition rates of 72% and 100%, respectively, suggesting a great application prospect of calamus and its EO on the behavioral control strategies of *P. xylostella*.

## 1. Introduction

The global application of synthetic insecticides for controlling agricultural pests has led to ever-increasing environmental and ecological issues, posing a severe threat to human health and ecosystem balance [1]. Multiple alternative eco-friendly strategies for pest management, including the application of biopesticides, from microbial to botanical biopesticides, have been sought and put into practice for many years [2,3,4]. Botanical insecticides from plant-derived natural compounds have indicated their enormous potential as an alternative for pest control and crop protection [5,6,7,8].

Behavioral manipulation based on the olfactory perception of pests is a green control technology that specifically regulates the behavior of pests, including sexual attractants, food attractants, repellents, and mating disruption [9,10]. Phytophagous insects usually utilize plant secondary metabolites to orient suitable host plants for foraging and oviposition. Plant volatiles additionally and significantly affect mating behavior, although species-specific sex pheromones commonly mediate the process [11]. The combination of host plant volatiles and sex pheromones can significantly increase the attraction effect on male insects and promote mating [12,13]. However, negative olfactory signals, such as those from non-host plants, usually have a negative effect on insect behavior [14,15,16]. For instance, the presence of non-host volatiles (NHVs) can impede male moth responses to the female sex pheromone by masking or inhibiting their effects, thereby resulting in mating disruption and further decreasing the population of the next generation of herbivorous insects [17,18,19,20]. Therefore, intercropping or relay cropping between host and non-host plants significantly affects the population of pests in various agroecosystems [21,22,23,24].

Essential oils (EOs) are steam-volatile or organic-solvent extracts of secondary metabolites derived from aromatic plants and are easily obtained through extraction from plant organs [25]. They are complex mixtures primarily consisting of volatile terpene hydrocarbons (monoterpenes and sesquiterpenes) and aromatic and oxygenated compounds, which dominate the aromatic characterization of plant materials [26]. Many plant EOs are well-documented for exerting efficient biological activities on herbivorous insects, including insecticidal, repellent, antifeedant, and growth regulatory activities [27,28]. Due to their cost-effectiveness, low toxicity to mammals, and high degradation characteristics, EOs have been applied to integrated pest management (IPM) [29,30,31,32].

The diamondback moth *Plutella xylostella* (Linnaeus) (Lepidoptera: Plutellidae) is a global pest of cruciferous plants (Brassicaceae), including various economically important food crops such as cabbage, cauliflower, and rapeseed. The yearly cost of the pest management of *P. xylostella* is huge and is causing enormous economic losses globally [33]. With the extensive application of insecticides, *P. xylostella* has developed broad-spectrum resistance and cross-resistance to various insecticides and further expanded with global climate change, which has led to control failures by insecticides [33,34,35,36,37]. In recent years, botanical extracts and EO applications have been considered eco-friendly and practical approaches for controlling the diamondback moth [38,39]. Many studies have focused on the insecticidal properties and antifeeding activities of larvae and oviposition deterrent effects on female moths. For example, the EOs from *Artemisia lavandulaefolia*, *Acorus calamus*, *Cedrus deodara*, *Murraya koenigii*, *Ocimum basilicum,* and *Pelargonium graveolens* have high insecticidal and repellent activities and strongly inhibit the feeding and growth of larvae [40,41,42]. Additionally, the extracts of *Ageratum conyzoides*, *Rosmarinus officinalis*, *Mentha piperita,* and *Datura stramonium* were reported to strongly inhibit the egg-laying of female moths [43,44,45]. However, the effect of EOs on sex pheromone communication during the courtship process of adults has been less studied.

Here, we studied the inhibition effect of seven non-host plant EOs on the reproductive-related behaviors of both male and female moths. Specifically, we tested the olfactory response of the antenna by performing electroantennogram recordings (EAGs) against the EOs of citronella, calamus, chamomile, lemon, sweet orange, chenpi, and tangerine, which have been reported to have insecticidal effects on the larvae of *P. xylostella* or other pests [42,46,47]. Then, laboratory behavioral assays and field traps were conducted to test the orientation of male moths to the combination of EOs and a synthetic female sex pheromone blend. The ternary blend consisted of (Z)-11-hexadecenal, (Z)-11-hexadecenyl acetate, and (Z)-11-hexadecenol in a 7:3:1 ratio, which was considered to be the most attractive to the male moths in the Chinese *P. xylostella* population [48]. Furthermore, the leaf discs and cage assays of oviposition were performed to study the effect of the EOs on inhibiting the egg-laying of females. This study deepens our understanding of how EOs work in the behavioral control of *P. xylostella* and provides a scientific basis for the application of EOs in green pest management.

## 2. Materials and Methods

### 2.1. Chemicals

The synthetic sex pheromone components of *P. xylostella* used in this study were (Z)-11-hexadecenyl acetate (Z11-16: OAc; CAS: 34010-21-4), (Z)-11-hexadecenal (Z11-16: Ald; CAS: 53939-28-9), and (Z)-11-hexadecen-1-ol (Z11-16: OH; CAS: 56683-54-6) (Shin-Etsu Chemical Co., Ltd., Tokyo, Japan). All the components were of analytical grade (>97%) and used without further purification. All the EO products (purity > 98%) were purchased from Guoguang Spice Factory (Ji’an, China). Detailed information of each EO, including the Latin name of the source plant and its taxonomy, the extracted plant tissues, the extraction methods, and the chemical abstracts service (CAS) numbers are listed in Table 1.

### 2.2. Insects

The experimental individuals of *P. xylostella* moths came from a laboratory colony, which was captured from various cabbage fields in Ningbo (Zhejiang Province, China) and were raised for at least three generations under artificial conditions with a 14L: 10D photoperiod at 25 ± 2 °C and 70% relative humidity. The insects were reared according to the methods of previous studies with some modifications as follows [49,50]. Briefly, the larvae were reared on Brassica chinensis plants within large gauze cages (1 × 1 × 1 m). After pupation, the males and females were separated and transferred to smaller cages (0.5 × 0.5 × 1 m). Newly emerged adults were individually placed into a test tube and fed with 10% (*v*/*v*) honey water on a cotton ball.

### 2.3. Electroantennogram Recordings (EAGs)

The EAGs were performed to assess the olfactory responses of male, virgin female, and gravid female *P. xylostella* moths to the EOs derived from seven non-host plants. The EOs were dissolved in paraffin oil and diluted to 200 μg/μL. A 10 μL solution was added to a filter paper (5 cm × 1 cm) and put into a Pasteur pipette for testing. Before testing, all the moths were transferred into the EAG laboratory to adapt to the new surrounding conditions, such as the room temperature and humidity, for approximately 1 h. The antennae were carefully excised at the base, trimmed slightly at both ends, and then fixed onto an EAG electrode coated with conductive glue. The electrode was positioned on an MP-15 micromanipulator (Syntech, Buchenbach, Germany) with a steel mesh screen connected to an electrical outlet ground. The amplified signal was captured using an Intelligent Data Acquisition Controller IDAC-2 (Syntech, Buchenbach, Germany) linked to a computer system. A stimulus controller (CS-55; Syntech, Buchenbach, Germany) delivered constant humidified (70%) air at a flow rate of 5 mL/s over the antennae through a Teflon tube with a diameter of 7 mm. For each antenna, the EAG response to each compound was calculated by averaging the values of three mechanical repetitions. Meanwhile, each antenna was tested against a green leaf volatile compound (Z)-3-hexene-1-ol as a reference odor to calibrate the individual differences in the response values. Six antennae from different moths were tested for biological duplication. The relative values to the EO or paraffin oil were calculated according to the following formula: relative value = (EAG response value to EO or paraffin oil)/(EAG response value to reference odor).

### 2.4. Bidirectional Selection Olfactometer Assays

The behavioral experiments were performed in a two-choice (Y-tube) olfactometer (3 cm in diameter) with two side arms with a length of 14 cm and one main arm with a length of 15 cm. The experiments were carried out under diffused light conditions (53 lux) from the top of the olfactometer. A charcoal-filtered airstream, humidified by passing through a wash bottle containing 50 mL of distilled water, was split into each arm of the olfactometer. Each arm had two *B. chinensis* leaf discs (1 cm in diameter) placed on wet filter paper as background stimuli to establish the context-based attraction. The synthetic sex pheromone blend (Z11-16: Ald, Z11-16: OAc, and Z11-16: OH with a ratio of 7:3:1 (*w*/*w*)) was prepared at a concentration of 10 μg/μL using n-hexane as the solvent, and then 5 μL of the solution was introduced into black rubber septa, which carried 50 μg of the sex pheromone blend (SP blend). Each EO was diluted to 200 μg/μL with n-hexane as a solvent. Then, 300 μL was applied to the black rubber septa above the carrying synthetic sex pheromone blend (SP blend + EO). The rubber lures were sealed in a polythene bag and stored at −20 °C. In the experimental setup, the test samples and controls were placed in odor tubes connected to the two sides of the Y-tube. One male moth (2-day-old) was introduced at the entrance of the main arm for behavioral observation. A selective response by the moths was defined as entering one-third of the tube wall and remaining for more than 5 s. After each test, the olfactometer was thoroughly cleaned using alcohol and distilled water before being dried in an oven at 120 °C. A total of fifty male moths were tested in each treatment. The selection percentage was calculated using the following formula: selection percentage = (number of moths in the selected odorant arm or control arm)/(total number of moths in the odorant arm + control arm) × 100%.

### 2.5. Field Traps

The field experiments were conducted in pesticide-free vegetable plots spanning a 13,320 square meter area located in Luotuo Town, Ningbo City, Zhejiang Province (coordinates: 121°32′43″ E, 29°56′22″ N; elevation: 4 m). The site comprised over 20 plastic greenhouses dedicated to cultivating cruciferous vegetables, such as pakchoi and green vegetables, year-round. Basin traps were used in the field assays. Each trap was comprised of a 20 cm basin, which was filled with a solution of 3% soapy water to submerge the captured moths, and a black trap core suspended using a wire 2 cm above the water surface. The cores of each EO were subjected to six different tests: SP, SP combined with different dosages of EO (1, 10, 50, and 200 mg), and a blank control. Each treatment was replicated four times. The traps were strategically positioned at intervals of 10 m apart, and the number of trapped individuals was recorded daily. Following each assessment, the captures were removed from the traps, and the trap positions were altered to minimize potential experimental errors arising from location variations.

### 2.6. Leaf Disc Assays of Oviposition

The oviposition preference experiments were conducted using the leaf disc assay. A total of four experimental groups were designed, with each group exposed to varying doses of EO (1, 100, 500, and 2000 μg), with six replicates per experiment. For each set of experiments, eight treatments were implemented by placing eight *B. chinensis* leaves (approximately 5 × 5 cm in size) on round plastic petri dishes with a diameter of 15 cm and a height of 2 cm. One leaf served as the control (CK, hexane), while the remaining seven leaves were evenly coated with all seven EOs. Two pairs of *P. xylostella* moths (two females and two males, 2-day-old moths) were introduced into each plastic petri dish. A small plastic cup containing a 10% sugar solution on cotton was placed at the center of each plastic petri dish to provide a food source. Two days later, the number of eggs per plant was quantified, and the investigation lasted six days. Each treatment was replicated four times. The statistical differences among the treatments were analyzed using a one-way ANOVA followed by LSD and indicated by different letters. The oviposition inhibition rate was calculated using the formula: inhibition rate = (number of eggs in the control group − number of eggs in EO treated group)/(number of eggs in the control group + number of eggs in EO treated group) × 100%.

### 2.7. Cage Assays of Oviposition

The cage assays were conducted to study the effect of seven EOs on the oviposition preferences of females under natural conditions. Six pots of B. chinensis seedlings with 2–3 true leaves were arranged in each screened-in cage (1 × 1 × 1 m), where three pots were equipped with black rubber septa carrying EOs and the other three with rubbers carrying the solvent. Three replicates were performed for each EO treatment. The prepared EO was carefully administered into the black rubber septa of the treatment group and securely fastened to the middle of the seedlings using wood strips and iron wires. At the same time, n-hexane was employed as a control substance. The pots containing the treatment and control groups were strategically positioned at opposite ends of the enclosure. Each cage was populated with five pairs of *P. xylostella* moths (five females and five males, 2-day-old moths). A small plastic cup containing a 10% sugar solution on cotton was positioned at the center of each cage to provide a food source. Two days following moth release, the number of eggs per plant was quantified, and this investigation continued for four consecutive days. The oviposition inhibition rate was calculated using the formula: inhibition rate = (number of eggs in the control group − number of eggs in EO treated group)/(number of eggs in the control group + number of eggs in EO treated group) × 100%.

### 2.8. Data Analysis

The differences in the EAG values between the EOs and sex pheromone compounds were analyzed using a one-way ANOVA followed by the least-significant difference (LSD) test. The same analysis method was used for the field traps among the different EOs and female oviposition dosages in the leaf disc assays. A Chi-square test was used to compare the choice ratio between the two sides for the Y-tube olfactometer assays. For the cage assays of oviposition, the statistical differences between each EO treatment and its control group were analyzed using a paired sample *t*-test. For the leaf disc assays of oviposition, the data were transformed using the relative amount of spawning to ensure homogeneity of variance. Specifically, the cos transformation was applied for the group treated with 1 μg of EO, while the groups treated with 100, 500, and 2000 μg underwent a log10 transformation. The data analysis was conducted using IBM SPSS Statistics 26. The figures were generated using GraphPad Prism 9.

## 3. Results

### 3.1. Electroantennographic (EAG) Responses of the Diamondback Moths to Seven Non-Host Plant Essential Oils

To estimate the olfactory perception of *P. xylostella* moths to seven EOs, we performed EAG assays on the antennae of both sexes. For the male moths, it was shown that the antennal relative values (mean ± SE) to citronella (1.26 ± 0.20), calamus (0.94 ± 0.12), sweet orange (1.00 ± 0.08), chamomile (0.87 ± 0.28), and tangerine (0.87 ± 0.18) EOs were significantly higher compared with the paraffin oil (CK) (0.49 ± 0.05), lemon (0.56 ± 0.01), and chenpi (0.69 ± 0.10) (Figure 1A). There was no significant difference between the lemon with chenpi EO and the same as the lemon with CK sample. These results suggest that the male moths have stronger abilities for sensing the volatiles of five EOs: citronella, calamus, sweet orange, chamomile, and tangerine. In the virgin female moths, the calamus EO elicited the strongest EAG values (0.99 ± 0.26) compared with any other ones. (Figure 1B). Similarly, significantly higher responses were observed when tested against the calamus EO (2.05 ± 0.59) in the gravid female moths (Figure 1C). On the other hand, the overall response values in the antennal of gravid female moths were higher than those in the virgin females, indicating that gravid females are more sensitive to sensing plant volatiles. 

### 3.2. Behavioral Responses of Male Moths to Sex Pheromones by Combining Essential Oils

To investigate the role of the seven non-host plant EOs in the mate location process of the male moths, we first tested the olfactory selection preferences of the male moths to either a single synthetic sex pheromone or a combination of pheromones and one EO in a bidirectional selection olfactometer. The results showed that significantly more (81% and 79%) individuals had a preference for the side that connected the air stream of a single synthetic sex pheromone rather than that of a combination with the citronella (*p* < 0.001) or calamus (*p* < 0.001) EOs (Figure 2). On the contrary, significantly fewer male moths (31%) selected the side of the synthetic sex pheromone blend rather than the combination of sex pheromones with the chamomile EO (*p* = 0.034). The combinations of sex pheromone blends with the other four EOs had no effect on the selection tendency. These findings suggest that both the citronella and calamus EOs inhibited the preference of male moths for the sex pheromone blend, while the chamomile EO had a synergistic effect on the attraction of the sex pheromone blend.

Field traps were further conducted to investigate the effect of seven EOs combined with synthetic sex pheromones on the orientation of male moths. It was shown that almost all of the EOs inhibited the trapping effect of the sex pheromone lures (Figure 3). In particular, the citronella (Figure 3A) and calamus EOs (Figure 3D) significantly decreased the captures of the sex pheromone trap at a very low concentration dosage (1 mg). Further, the orientation of male *P. xylostella* moths to the sex pheromones was significantly inhibited by all the essential oils at 10 mg, with calamus essential oil exhibiting the highest inhibition rate of 72%. The inhibition rate of all the essential oils consistently decreased as the dosage increased. These findings suggest that the tested essential oils act as antagonists for sex pheromone sensing and disrupt the orientation of the male moths to the sex pheromone lures.

### 3.3. Oviposition Preference of Female Moths to Seven Plant Essential Oils

Multiple-choice tests were first performed to study the oviposition preference of the gravid female moths for the seven essential oils in different doses. The results showed that the application of essential oils made a prominent impact on the yield of eggs (Figure 4). Treatment with a low dosage of the tangerine essential oil significantly increased the amount of egg-laying (Figure 4A). Conversely, the essential oils significantly inhibited the spawning number as the dosage increased. Specifically, the application of the calamus and chamomile EOs (100 μg) significantly reduced the amount of egg-laying (Figure 4B), and with the dosage increasing, all seven essential oils significantly inhibited the yield of eggs. Notably, the calamus EO at 500 μg and 2000 μg showed the highest inhibitory effect with a 100% inhibition rate. To further investigate the effect of the essential oils on oviposition under natural conditions, the egg-laying preference assays on host plants, which were treated with EOs and the solvent (control group), were conducted in large cages. The results showed that the female moths laid significantly fewer (*p* = 0.006) eggs on plants treated with the calamus EO at 20 mg compared to the control plants, with an inhibition rate up to 63.16% (Figure 5B). On the contrary, the number of eggs on the plants treated with the sweet orange EO was significantly higher (*p* = 0.021) than that of the control plants, increasing the egg production by 50%. There was no significant difference between the groups treated with the other five EOs and solvents. At the dosage of 2 mg, no statistical difference was found between any of the EO treatments and the control, although the oviposition inhibition rate of the calamus EO was 100% (Figure 5A). 

## 4. Discussion

The volatile secondary metabolites of plants are considered to play an essential role in modulating the behavior of insects, such as courtship, mate location, and oviposition, thereby influencing the population quantity and survival rate of the offspring [11]. Insects detect and discriminate volatile compounds in the environment using their highly developed olfactory system, and it has been proven that the antennal response spectra to their ecologically relevant volatiles are species-specific [51]. In this study, our results demonstrated that both male and female moths of *P. xylostella* can specifically sense EOs from non-host plants, especially the calamus EO. The calamus EO elicited a robust EAG value in male moths and the strongest response in either the virgin or gravid females, implying that the calamus EO may have significant effects on the behavior of both sexes. Higher response values in the gravid female moths compared with virgin ones suggest that females have a stronger ability to sense plant volatiles, which may relate to the quality of the spawning site. 

Further laboratory behavioral assays combined with field traps suggested that the citronella and calamus EOs strongly inhibited the orientation of male moths to synthetic sex pheromone blends. In the field traps, although all EOs at higher dosages significantly reduced the number of male moths captured using the synthetic sex pheromone blend, only the citronella and calamus EOs at low dosages had a significant inhibitory effect on sex pheromone orientation, which is consistent with the results of the laboratory behavioral assays. These studies suggested that the two EOs can be used in pest behavior control via mating disruption. In the mating behavior of insects, plant volatiles are considered to additionally and significantly promote successful mating behavior, although species-specific sex pheromones are usually the principal cues for mate location [11]. However, adverse olfactory signals from unsuitable plants usually lead to a negative behavioral response. It has been reported that herbivore-induced cotton volatiles strongly suppress the orientation of the moth *Spodoptera littoralis* to host plants and mates by suppressing the olfactory sensitivity of male moths to the main pheromone component [52]. Over the past few decades, mating disruption has been one of the most successful approaches for green pest management, and the application of plant volatiles is considered to be an alternative strategy [49,53]. 

In female moths, by performing multiple-choice experiments in leaf discs, we found that all high dosages of EOs significantly inhibited the yield of eggs. The calamus and chamomile EOs significantly decreased the number of eggs under a low dosage, suggesting higher biological activities of the two EOs. Further cage assays under natural conditions suggested that the calamus EO strongly inhibited the oviposition of female moths, with a maximum inhibition rate of up to 100%, suggesting its high activity on deterred oviposition. The behavioral assays in both male and female moths indicated that the citronella and calamus EOs have great potential to be applied in the behavioral control of the integrated pest management of *P. xylostella*. The citronella EO has been reported to effectively control the population of *P. xylostella* in cabbage by inhibiting the development of larvae and repelling the oviposition of female moths, with the highest level of 100% [46]. However, the effect on sex pheromone communication has not been studied yet. According to our results in this study, the effect of the citronella EO on mating disruption also contributes to the suppression of the pest population. The inhibition effect of the calamus EO from *A. gramineus* on reproductive behaviors in both sexes has not been reported. An EO from another closely related species, A. calamus, has been documented as an effective repellent on the oviposition of *P. xylostella* female moths [46,54]. 

It is worth noting that the results of behavioral assays were conflicting between the laboratory and natural conditions. The chamomile EO had a synergistic effect on the attraction of sex pheromone blends to the male moths in our laboratory behavioral assays, which was contrary to the results of the field traps. In the oviposition assays, the sweet orange EO at dosages of 500 and 2000 μg significantly inhibited egg-laying in the leaf disc assays. However, 2 mg of the sweet orange EO did not affect the oviposition, and 20 mg of this EO significantly increased the egg-laying amount by 50% in the cage assays. A probable reason may be that the complex background odor in the field modified the effect of specific volatile components, the representation of which also altered as the concentration increased. It was revealed that the mixture of odors present in the environment influences the olfactory ability of a moth, and the representation of specific odors can be altered by the background odors [55,56]. Studies on the tobacco hornworm *Manduca sexta* have shown that its olfactory ability to trace the *Datura wrightii* flower bouquet can be excited or inhibited depending on the background odors by influencing the neuronal representation in the central olfactory system of the moth [57]. 

## 5. Conclusions

Both the male and female moths of *P. xylostella* can specifically perceive the volatile signals from non-host plant EOs. A low dosage of the calamus EO from *A. gramineus* inhibited the attraction of synthetic sex pheromones to male moths, with the highest inhibition rate of up to 72%, and reduced the egg-laying amount of female moths on host plants, with the highest inhibition rate of 100%. These results suggest a high potential application of the calamus EO on the behavioral control of *P. xylostella* via mating disruption and oviposition inhibition. Moreover, *A. gramineus* plants, which are widely cultivated in China as a medicinal herb, have the potential to act as repellent plants in “pull-push” strategies of integrated pest management. Next, we will identify the key bioactive components in the calamus EO and further study the olfactory mechanism in regulating mating and egg-laying behaviors. Additionally, further study is needed on the effectiveness of calamus oil and its bioactive components for controlling the population of *P. xylostella* in the field.

## Figures and Tables

**Figure 1 insects-15-00227-f001:**
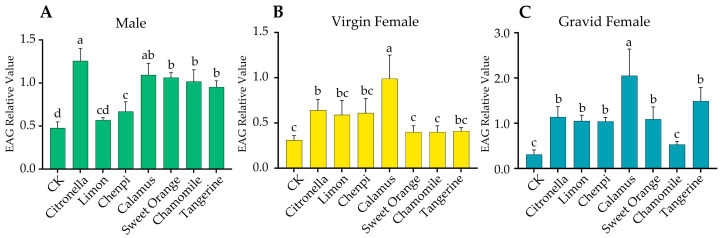
EAG relative responses of the male (**A**), virgin female (**B**), and gravid female (**C**) moths of *P. xylostella* to seven essential oils of non-host plants. The statistical differences among the treatments were analyzed using a one-way ANOVA followed by LSD and indicated by different letters: the male moths (*p* < 0.001), the virgin female moths (*p* < 0.001), and the gravid female moths (*p* < 0.001). The control groups (CK) were treated with a paraffin oil solvent.

**Figure 2 insects-15-00227-f002:**
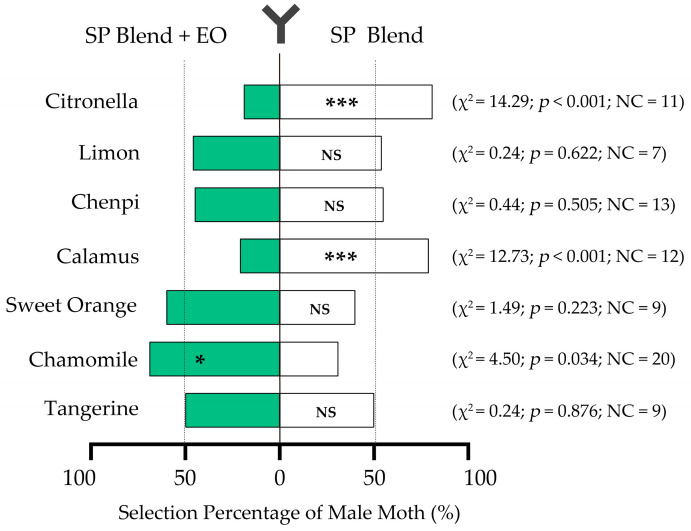
Selection preferences of the male moths to the synthetic sex pheromone (SP) blend of *P. xylostella* and the combination with EOs. A Chi-square test was used to compare the choice ratio between the two sides. The asterisk (*) indicates that the difference is significant at the 0.05 or 0.001 level (*, *p* < 0.05; ***, *p* < 0.001), and NS indicates no significant differences between the two sides. NC indicates that the tested moths had no choice. Fifty male moths were tested in each treatment.

**Figure 3 insects-15-00227-f003:**
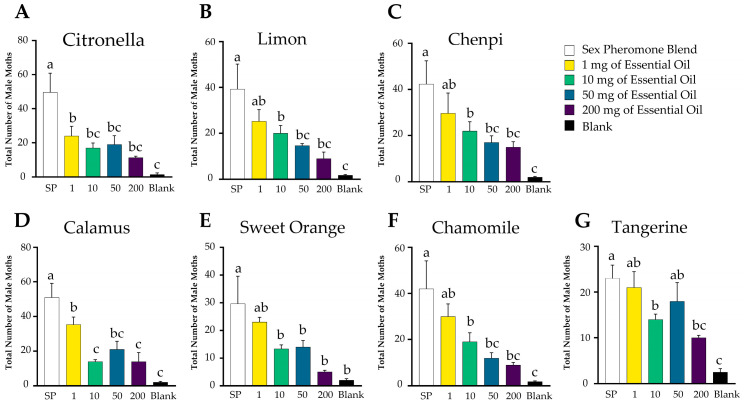
Field traps of *P. xylostella* by sex pheromone combined with different essential oils: (**A**) citronella; (**B**) lemon; (**C**) chenpi; (**D**) calamus; (**E**) sweet orange; (**F**) chamomile; (**G**) tangerine. SP indicates a synthetic sex pheromone blend of the female moth and SP + EO indicates the combination of a sex pheromone blend with an essential oil. Each treatment was replicated four times. The statistical difference among the treatments was analyzed using a one-way ANOVA followed by LSD, and indicated by different letters. citronella: *p* = 0.002; lemon: *p* = 0.004; chenpi: *p* = 0.004; calamus: *p* < 0.001; sweet orange: *p* = 0.005; chamomile: *p* = 0.005; tangerine: *p* = 0.001.

**Figure 4 insects-15-00227-f004:**
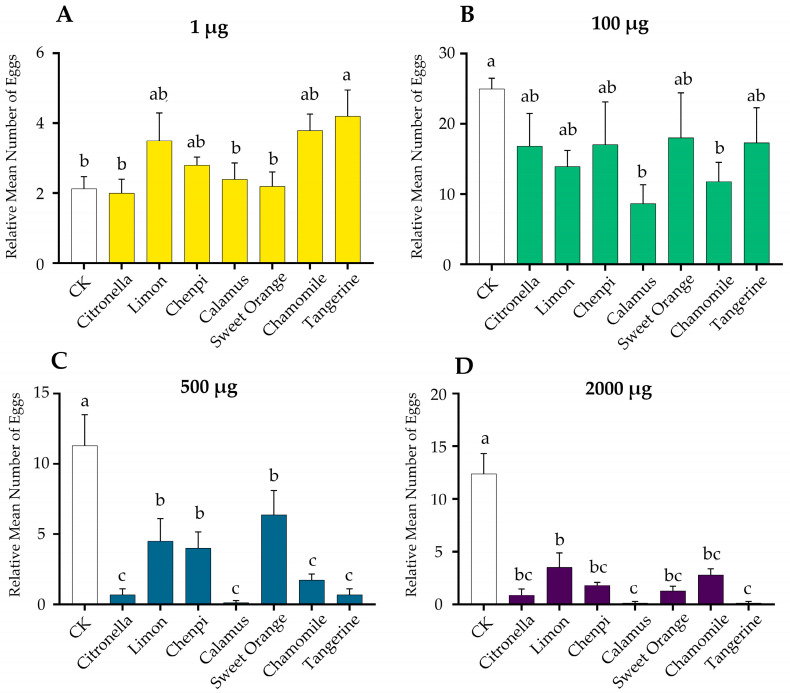
Leaf disc assays of the oviposition preferences of gravid female moths to 1 μg (**A**), 100 μg (**B**), 500 μg (**C**), and 2000 μg (**D**) of the seven EOs. Each treatment was replicated four times. The statistical differences among the treatments of each group were analyzed using a one-way ANOVA followed by LSD, and indicated by different letters.

**Figure 5 insects-15-00227-f005:**
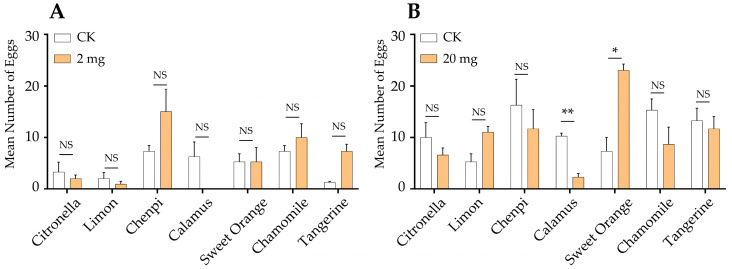
Cage assays of the oviposition preference to 2 mg (**A**) and 20 mg (**B**) EOs. Each treatment was replicated four times. The differences between the EO treatment and its control group were analyzed using a paired sample *t*-test. The asterisks indicate that the difference is significant at the 0.05 or 0.01 level (*, *p* = 0.021; **, *p* = 0.006), and NS indicates no significant difference.

**Table 1 insects-15-00227-t001:** Seven essential oils from the non-host plants for the bioassays.

Essential Oil	Plant Tissue	Extraction Method	Taxonomy	CAS NO.
*Cymbopogon citratus* (Citronella)	Whole Plant	Simultaneous Distillation Extraction	*Cymbopogon* (Gramineae)	8000-29-1
*Citrus limon* (Limon)	Pericarp	Cold-pressed	*Citrus* (Rutaceae)	8008-56-8
*C. reticulata* (Pericarpium Citri Reticulatae, Chenpi)	Pericarp	Cold-pressed	*Citrus* (Rutaceae)	_
*A. gramineus* (Calamus)	Whole Plant	Simultaneous Distillation Extraction	*Acorus* (Araceae)	91745-11-8
*C. sinensis* (Sweet Orange)	Pericarp	Cold-pressed	*Citrus* (Rutaceae)	8008-57-9
*Matricaria chamomilla* (Chamomile)	Inflorescence	Simultaneous Distillation Extraction	*Matricaria* (Compositae)	8015-92-7
*C. reticulata* (Tangerine)	Pericarp	Cold-pressed	*Citrus* (Rutaceae)	8008-31-9

## Data Availability

All the data and resources generated for this study are included in the article.

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
