# Peer review of "Inhibition Effect of Non-Host Plant Volatile Extracts on Reproductive Behaviors in the Diamondback Moth Plutella xylostella (Linnaeus)"

_insects, 2024, doi:10.3390/insects15040227_

Round 1
Reviewer 1 Report
Comments and Suggestions for Authors
Dear authors, see suggestions inside of document

Reviewer 2 Report
Comments and Suggestions for Authors
In my opinion, the present study on the effect of seven essential oils on the attraction of the pest Plutella xylostella is interesting. In view of the results, the work is publishable and with important results. The introduction should be improved, especially at the background level and to better justify the need for the study, as well as to indicate the existing bibliography on the effect of the EOs studied. It should also be improved methodologically by including some missing details, such as the justification for the use of these chemical synthetics of pheromone attractants or the origin of the essential oils. In addition, some details on preparations, calculations or units of measurement, as well as statistics. The quality and visibility of the figures should be improved, as included in the attached files, as should some of the details of the legends. The discussion should be improved, including more background and analysis of the results obtained. Its viability should be projected in the possible control strategies both at the infrastructure level and also at the economic level in the extraction of essential oils and its commercial projection, if possible. All this should also be included in the conclusions, highlighting, as has already been done, Calamus as the oil with the greatest activity. The doses used should also be justified, and why these doses were used and not others. Plese, revise also formatting. Further comments are included in the attached document.

In my opinion, a moderate edition of English should be carried out and checked for plagiarism.
Round 2
Reviewer 2 Report
Comments and Suggestions for Authors
In my opinion, this second version of the manuscript has been significantly improved. The issues I raised have been almost completely addressed. I think the article is publishable. A few minor points:
- In the summary and abstract, I would include the differences observed in the quantitative data. For example, in 'strongly', 'significantly decreased', 'significantly inhibited', 'inhibited' or 'reduced'.
- I think 'cruciferous plants' is lower case.
- In the introduction, add a space in 'moth[38,39]'. Please check the entire manuscript for formatting. In the graphics, although it is a bit tedious, the font should be standardised to 'Palatino linotype'.
- In Table 1, I think that bibliographic references should be given and/or the extraction protocols should be briefly described, although this may seem obvious.
- On a background level, I think it can be improved, although I think it can be published as is.
- Open a new section '5. Conclusions' to include the conclusions. Instead of repeating what you have done, state the main conclusions and projections of the study.
- Open a new section '6. References'. Also check the bibliography at the format level.
Comments on the Quality of English LanguageIn my opinion, English is fine.
Author Response
Dear Reviewer,
Thanks for your insightful valuable suggestions again. We made minor revisions to the manuscript according to your comments. Please see the latest version.
The point-by-point response to your comments is as follows:
- In the summary and abstract, I would include the differences observed in the quantitative data. For example, in 'strongly', 'significantly decreased', 'significantly inhibited', 'inhibited' or 'reduced'.
Response: We improved the summary and abstract with quantitative data.
Simple Summary: ......The results demonstrated that 10 mg of calamus (Acorus gramineus) and citronella (Cymbopogon citratus) EOs reduced the attraction of synthetic sex pheromones to male moths up to 72% and 66%, in a sensitive way. Calamus EO also decreased the egg-laying amount of female moths on host plants with the highest inhibition rate of 100%.......
Abstract:......Notably, calamus (Acorus gramineus) EO inhibited the preference of male moths for synthetic sex pheromone blend and reduced the egg-laying number of female moths on host plants, with the highest inhibition rates of 72% and 100% respectively......
- I think 'cruciferous plants' is lower case.
Response: Corrected.
- In the introduction, add a space in 'moth[38,39]'. Please check the entire manuscript for formatting. In the graphics, although it is a bit tedious, the font should be standardised to 'Palatino linotype'.
Response: The format of the entire manuscript was checked and modified. We also modified the font to 'Palatino linotype' in all the figures.
- In Table 1, I think that bibliographic references should be given and/or the extraction protocols should be briefly described, although this may seem obvious.
Response: Thank you for your valuable suggestions. We have tried to obtain the detail extraction protocols from the manufacturer (Guoguang Spice Factory) but were rejected. To provide more information about the EO productions, we added CAS numbers to Table 1.
- On a background level, I think it can be improved, although I think it can be published as is.
Response: Thank you for your valuable suggestions. We have modified some phrases to make the sentence easier to understand.
- Open a new section '5. Conclusions' to include the conclusions. Instead of repeating what you have done, state the main conclusions and projections of the study.
Response: Modified.
- Open a new section '6. References'. Also check the bibliography at the format level.
Response: We have checked dozens of the latest published papers in this journal, none of them made a new section for 'References'. So we kept it in the original form. We have checked the format of all references carefully.